Changes and sex- and age-related differences in the expression of drug metabolizing enzymes in a KRAS-mutant mouse model of lung cancer

Li Xiaoyan 1
Lu Yiyan 1
Ou Xiaojun 1
Zeng Sijing 1
Wang Ying 1
Qi Xiaoxiao 1
Zhu Lijun zhulijun@gzucm.edu.cn 1
Liu Zhongqiu liuzq@gzucm.edu.cn 1 2
1 International Institute for Translational Chinese Medicine, Guangzhou University of Chinese Medicine , Guangzhou , China
2 State Key Laboratory of Quality Research in Chinese Medicine, Macau University of Science and Technology , Macau , China
Silva Pedro
Electronic publication date: 2020 Nov 18
Publication date: 2020
Volume: 8
Electronic Location ID: e10182
Received 2020 Mar 2; Accepted 2020 Sep 23
Copyright: ©2020 Li et al.
Copyright year: 2020
Copyright holder: Li et al.
License: This is an open access article distributed under the terms of the Creative Commons Attribution License, which permits unrestricted use, distribution, reproduction and adaptation in any medium and for any purpose provided that it is properly attributed. For attribution, the original author(s), title, publication source (PeerJ) and either DOI or URL of the article must be cited.
License URL: https://creativecommons.org/licenses/by/4.0/

Keywords: Drug metabolizing enzyme (DME), Liquid chromatography tandem massspectrometry (LC-MS/MS), KRAS mutant mouse model of lung cancer (KRAS mice)

Funding: National Natural Science Foundation of China 81720108033 Nature Science Foundation of Guangdong Province 2015B020233015 Guangdong Province Universities Colleges Pearl River Scholar Funded Scheme (2015 and 2016) This work was supported by the projects of the National Natural Science Foundation of China [81720108033], the Nature Science Foundation of Guangdong Province [2015B020233015], and the Guangdong Province Universities and Colleges Pearl River Scholar Funded Scheme (2015 and 2016). The funders had no role in study design, data collection and analysis, decision to publish, or preparation of the manuscript.

==============================
Background

This study aimed to systematically profile the alterations and sex- and age-related differences in the drug metabolizing enzymes (DMEs) in a KRAS-mutant mouse model of lung cancer (KRAS mice).

Methodology

In this study, the LC-MS/MS approach and a probe substrate method were used to detect the alterations in 21 isoforms of DMEs, as well as the enzymatic activities of five isoforms, respectively. Western blotting was applied to study the protein expression of four related receptors.

Results

The proteins contents of CYP2C29 and CYP3A11, were significantly downregulated in the livers of male KRAS mice at 26 weeks (3.7- and 4.4-fold, respectively, p < 0.05). SULT1A1 and SULT1D1 were upregulated by 1.8- to 7.0- fold at 20 (p = 0.015 and 0.017, respectively) and 26 weeks (p = 0.055 and 0.031, respectively). There were positive correlations between protein expression and enzyme activity for CYP2E1, UGT1A9, SULT1A1 and SULT1D1 (r2 ≥ 0.5, p < 0.001). Western blotting analysis revealed the downregulation of AHR, FXR and PPARα protein expression in male KRAS mice at 26 weeks. For sex-related differences, CYP2E1 was male-predominant and UGT1A2 was female-predominant in the kidney. UGT1A1 and UGT1A5 expression was female-predominant, whereas UGT2B1 exhibited male-predominant expression in liver tissue. For the tissue distribution of DMEs, 21 subtypes of DMEs were all expressed in liver tissue. In the intestine, the expression levels of CYP2C29, CYP27A1, UGT1A2, 1A5, 1A6a, 1A9, 2B1, 2B5 and 2B36 were under the limitation of quantification. The subtypes of CYP7A1, 1B1, 2E1 and UGT1A1, 2A3, 2B34 were detected in kidney tissue.

Conclusions

This study, for the first time, unveils the variations and sex- and age-related differences in DMEs in C57 BL/6 (WT) mice and KRAS mice.

Introduction

Many studies have posited that disease states, as well as sex- and age-related differences can alter the expression of drug metabolizing enzymes (DMEs), and in turn change the metabolism and detoxification of drugs by remodeling the hepatic absorption, distribution, metabolism and excretion (ADME) (Court, 2010; Li et al., 2017; Nyagode et al., 2014; Xu et al., 2019; Wu & Lin, 2018). Therefore, we specifically investigated the changes in DME expression levels in a disease model with age- and sex-related differences.

The KRAS mutation, the most frequently mutated isoform of RAS, accounts for > 85% of RAS-driven cancers (Ding et al., 2008; Nussinov et al., 2015). However, to date, it is still a major challenge to develop novel drugs that effectively treat KRAS mutant lung cancer (Mccormick, 2015). Considering the widespread and incurable nature of this disease, the metabolic profile of drugs is urgently needed to determine when KRAS is mutated. The mouse genome is 99% identical to the human genome, and the organs and systemic physiology of mouse have similar patterns with humans (Ribeiro et al., 2016). Hence, mice have been widely used in current cancer research (∼59% of the total number of animals used) (Wang et al., 2020). In our study, a KRAS-mutant mouse model of lung cancer (KRAS mice, spontaneous tumors in the lung) was used to study the changes in DMEs in the development of lung cancer. KRAS mice were first observed to have small pleural nodules at one week of age, and numerous pleural lesions started to appear in 5-weeks-old mice (Johnson et al., 2001). Advanced tumours begin to appear in the lung of KRAS mice at 20 weeks and its life span is approximately 28 weeks (Johnson et al., 2001).

The activation and inactivation of exogenous drugs are mainly regulated by drug metabolizing enzymes (DMEs), including cytochrome P450s (CYPs), UDP-glucuronosyltransferases (UGTs), and sulfotransferases (SULTs) (Yan et al., 2014; Xie et al., 2017). The changes in DMEs can further affect the efficacy of the drug and even increase the side effects. For instance, irinotecan, which is used in the treatment of metastatic colorectal cancer, causes severe intestinal toxicity attributed to damage to UGT1A (Ribeiro et al., 2016). It was also reported that CYP3A has higher expression in osteosarcoma (By (Dhaini et al., 2003). The activation/inactivation of anticancer drugs metabolized by CYP3A would lead to changes in curative effect. Sex and age are important factors influencing the expression level of DMEs (Kennedy, 2008; Zheng et al., 2018). For age-related differences, a related report showed that the enzyme activity of UGT1A increases before 20 years of age and then decreases (Court, 2010). Sex-related differences characterize the metabolism of many drugs used frequently in the clinic (Waxman & Holloway, 2009). For instance, men showed a 38% faster clearance of olanzapine than women (Bigos et al., 2008). Therefore, a thorough understanding of the variations in DMEs is beneficial and indispensable for pharmacological evaluation and rational clinical drug use.

To effectively treat the KRAS-mutant lung cancer, researchers have used many drugs, such as gefitinib, erlotinib, cisplatin, trametinib, and pazopanib (Kim et al., 2018; Pujol et al., 2006). Changes in DMEs could alter the metabolic characteristics of these drugs, further affecting their efficacy in vivo. The drugs erlotinib and cisplatin are mainly metabolized by CYP3A4, an ortholog of mouse CYP3A11 (Hagleitner et al., 2015; Sanoff et al., 2010). Alterations in the activity of CYP3A4 could potentially have a pronounced effect on drug exposure. In other words, downregulation of CYP3A4 could reduce sorafenib hepatotoxicity (Yan et al., 2015). However, little is known about the alterations in DMEs after KRAS mutation, thus causing some difficulties in understanding the fate of drugs in vivo and leading to confusion about the efficacy and side effects.

MS-based quantifications are different from traditional immunogenic methods and can increase the sensitivity and high throughput of the absolute quantification of proteins. In our study, an LC-MS/MS method was employed to determine the protein expression of DMEs in WT and KRAS mice (Chen et al., 2017). In addition, the possible mechanism for the variations in DMEs was investigated. We intend to provide a valuable reference for pharmacological evaluation and rational clinical drug use in patients with KRAS mutated lung cancer.

Materials and Methods

Chemicals and Reagents

Ammonium bicarbonate (AB), dithiothreitol (DTT), iodoacetamide (IAA), trifluoroacetic acid (TFA) and phenylmethanesulfonyl fluoride (PMSF) were bought from Sigma-Aldrich, USA. Sequencing grade modified trypsin was obtained from Promega (Madison, WI). All peptides and internal standard (purity > 95%) were got from Your R&D Partner. HPLC-grade methanol, formic acid and acetonitrile were acquired from Merck (Darmstadt, Germany).

NADPH solution A and NADPH solution B were got from BD Bioscience, USA. Alamethicin, Tetracycline, Glucosyl monophosphate, Uridine diphosphate glucuronic acid (UDPGA), 3′-phosphoadenosine-5′-phosphosulfate (PAPS), MgCl2, Chlorzoxazone, Testosterone, 6β-hydroxytestosterone, Propofol, Dopamine, 6-hydroxy chlorzoxazone, Propofol-glucuronide and 4-Nitrophenyl sulfate metabolite were acquired from Sigma-Aldrich, USA. Dopamine 3-O-sulfate and dopamine 4-O-sulfate were got from TRC, Canada. P-Nitrophenol was bought from Aladdin, China. Genistein and ammonium acetate were got from Chengdu Mansite Biotechnology Co., Ltd. and Dalian Meilun Biotechnology Co., Ltd., respectively. Coomassie brilliant blue, providing for protein measurement, was bought from Bio-Rad (Hercules, California, USA).

Animals

Male and female C57 BL/6 mice (5, 10, 15, 20, 26 weeks, n = 5) were obtained from Vital River Laboratory Animal Technology Co. Ltd (Beijing, China). Male and female B6.129S-Krastm3Tyj/Nci (K-rasLA2) (5, 10, 15, 20, 26 weeks, n = 5) were acquired from National Cancer Insitute. The animals were kept at controlled temperature of 24−26 °C and humidity of 50–60%, with a 12 h light-dark cycle. The permission of all animal experiments was obtained from the Institutional Animal Care and Use Committee of the International Institute for Translational Chinese Medicine (IITCM_20171105). Before the experiment, animals were fasted overnight but allowed free access to water. All procedures were performed under diethyl ether anesthesia and the efforts were made to minimize suffering. After the animal experiment was completed, the animal bodies were frozen and sent to professionals for harmless treatment.

Histopathological analysis of lung tissues

Lung tissues were acquired from KRAS mice and wild-type (WT) mice. The morphology of lung tissues was observed under stereoscopic microscope (Leica, M165C), then tissues were fixed by 4% paraformaldehyde. After paraformaldehyde fixation and paraffin embedding, mouse lung tissues were sliced and stained with hematoxylin for 30 s and 0.5% eosin for 10 s, and covered with neutral gum. The images were obtained under microscopy (NIKON Eclipseci, Japan).

Preparation of mouse S9 fractions

Mouse tissue (liver, kidney and intestine) were isolated from WT and KRAS mice. The tissues were minced and washed with ice cold saline. Ice-cold homogenization buffer (50 mM potassium phosphate, 250 mM sucrose, 1 mM EDTA, PH 7.4) with 0.28 mM phenylmethylsulfonyl fluoride (PMSF) was added to the minced tissues and homogenized until an even suspension was obtained. Then the suspension was centrifuged at 9, 000 × g for 20 min at 4 °C. The supernatant was collected as S9 fractions (Tang et al., 2012; Zhu et al., 2010). Liver tissue was handled with a single sample of each mouse while kidney and intestine tissues of each group of mice (n = 5) were mixed into same sample. Protein concentrations of mouse S9 fractions were detected by coomassie brilliant blue and the bovine serum albumin was selected as the standard.

LC-MS/MS analysis

Eight isoforms of CYPs (CYP1B1, CYP2C29, CYP2D22, CYP2E1, CYP3A11, CYP3A25, CYP7A1 and CYP27A1), ten isoforms of UGTs (UGT1A1, UGT1A2, UGT1A5, UGT1A6a, UGT1A9, UGT2A3, UGT2B1, UGT2B5, UGT2B34 and UGT2B36) and three isoforms of SULTs (SULT1A1, SULT1B1 and SULT1D1) were analyzed. The methods of sample preparation and quantifying DMEs amounts by UHPLC/MS-MS were consistent with our previous study and dynamic MRM chromatograms of 21 subtypes were displayed in Fig. S1 (Chen et al., 2017). Samples were analyzed by using an Agilent 6490 triple quadruple mass spectrometer coupled with 1290 Infinite UHPLC system. A Poroshell C 18 column (2.1 mm ×100 nm, 2.7 µm; Agilent Technologies) was used for separation. In this study, the protein amounts of DMEs were represented in the form of pmol protein per S9 fraction protein (pmol/mg). The quantification of protein levels was performed two independent experiments. All samples were performed in triplicate in each independent experiment and data were presented as mean ± SD.

Enzyme assays of liver S9 fractions

Enzyme activities of CYP2E1, CYP3A11, UGT1A9, SULT1A1 and SULT1D1 were measured by specific probe substrates in vitro (chlorzoxazone, testosterone, propofol, p-nitrophenol and dopamine, respectively). The enzyme activities of these isoforms in mice were determined by incubating S9 fractions with appropriate substrate concentrations. Production of metabolites was quantified to value the activities of these isoforms between WT and KRAS mice at their different age. The incubation systems of CYPs, UGTs and SULTs, are in accordance with our previous articles with minor modification (Xie et al., 2017; Yan et al., 2015; Zheng et al., 2018). In order to terminate the reaction, 200 µL methanol with 200 nM genistein was added. Then, the solution was vortexed and thereafter centrifuged for 30 min at 18,000 g. Then the supernatant of all samples was injected to analyze by LC-MS/MS. The enzyme activity was measured from 2 independent experiments. Each sample was performed in triplicates in each independent experiment and data was presented as mean ± SD.

Western blotting

The protein levels of aryl hydrocarbon receptor (AHR), bile acid receptor (FXR), pregnane X receptor (PXR) and peroxisome proliferator-activated receptor (PPARα) were determined in male WT and KRAS mice at 26 weeks, and β-actin was used as an internal control. The S9 samples were mixed with 5 × loading buffer and the mixture was denatured at 100 °C for 5 min. An equal amount of protein (40 µg) was separated by SDS-PAGE at a voltage of 120 V to the correct band size and the protein was subsequently transferred from the gel to the PVDF membrane. Then, the membrane was blocked for 1 h with 5% non-fat milk (w/v) in Tris-buffered saline containing 0.1% Tween-20 (TBST). The corresponding primary antibodies, against mouse peroxisome proliferator-activated receptor (PPARα, sc-398394), pregnane X receptor (PXR, ab118336), bile acid receptor (FXR, ab28480) and aryl hydrocarbon receptor (AHR, ab2769) and β-ACTIN (from Cell Signaling Technology, CST, Boston, USA) were diluted to a recommended dilution of 1:1000 with 5% non-fat milk according to the manufacturer’s instructions. After blocking, the membrane was incubated with the corresponding primary antibodies at 4 °C overnight with gentle shaking and was then washed before being incubated with the corresponding secondary antibody at a dilution of 1:2000–1:3000 for 1 h at room temperature. ECL chemiluminescence was used to detect the signals and each protein band was quantified by Image J (National Institutes of Health, Hercules, CA, USA). The WB analysis was performed from 2 independent experiments, and each target protein was analyzed twice in each independent experiment. The data was presented as mean ± SD.

Data analysis

One-way ANOVA analysis, non-parametric test and independent sample t-tests were conducted using SPSS 19.0 to evaluate statistical differences. Correlation analyses were performed using SPSS 19.0 and GraphPad Prism 7, according to the Pearson product-moment correlation for normal related data and Spearman’s rank correlation for non-normally related data. Partial least squares discriminant analysis (PLS-DA) was performed to visualize the changes of DMEs after KRAS mutation using SIMCA-P 14.0 tool (Umetrics, Umea, Sweden). In each case, a value of p <0.05 denotes statistical significance for all of the statistical analyses.

Results

The phenotypic characteristics of KRAS mice

Pulmonary morphology was observed using a stereoscope (Figs. 1A–1B). Compared with those in the WT mice, many lung nodules were observed in the lung tissues of the KRAS mice (Fig. 1B, As the arrows point), and the lung tissues appeared dull overall. Histological and pathological features of the WT and KRAS mice were detected by H&E staining (Figs. 1C–1H and Fig. S2). The morphology of lung cells in the WT mice was normal, whereas in the KRAS mice, the lung cells were hyperproliferative (Figs. 1F–1H and Fig. S2, As the arrows point). The nuclei were deeply stained and lager.

Figure 1 Morphology and H&E staining of lung tissues from the WT and KRAS mice.

(A–B) Pulmonary morphology in the WT and KRAS mice under a stereoscopic mirror. The arrows pointed out the lung nodules in KRAS mice. (C–H) Histological and pathological features of the WT and KRAS mice were detected by H & E staining at 26 weeks. The arrows partly pointed out the hyperproliferative lung cells in KRAS mice.

Figure 2 Alterations in DME protein content with KRAS mutation in the different tissues at different ages.

(A–E) Comparative data analysis of DMEs protein content in the liver of the male WT and KRAS mice was performed using PLS-DA plot at different ages. The solid black dots represent the KRAS mice, and the open black dots represent the WT mice (n = 5). (F–H) The relative expression levels of DMEs in the liver, intestine and kidney tissue in the male mice at different ages. Protein levels in the male WT mice (n = 5) were normalized to those in the male KRAS mice (n = 5). The data were analyzed by independent sample t-tests (for normally distributed data) and Mann–Whitney U analysis (for non-normally distributed data). The symbol “*” indicates a displayed significant difference between the male WT and KRAS mice at the same age, p < 0.05.

Alterations in the protein contents of DMEs by KRAS mutation

In male mice, PLS-DA analysis was applied to evaluate the clustering between the male WT and KRAS mice based on the expression of 21 DMEs (Figs. 2A–2E). We observed obvious distinctions between the male WT and KRAS mice at 5, 10, 15, 20 and 26 weeks. These results demonstrated the differences in DMEs among them. As shown in Fig. 2H in liver tissue, SULT1A1 increased by 2.4-fold at 5 weeks (p = 0.016); CYP27A1 and UGT1A1 increased by 1.9-fold and 2.3-fold at 15 weeks respectively (p = 0.001 and p > 0.05, respectively); SULT1A1 and SULT1D1 were upregulated by 3.4-fold and 1.8-fold at 20 weeks, respectively (p = 0.015 and p = 0.017, respectively); and at 26 weeks, SULT1A1 and SULT1D1 were upregulated by 2.0-fold and 1.8-fold respectively (p > 0.05 and p = 0.031, respectively), and CYP2C29, CYP3A11, CYP27A1 and UGT1A5 decreased by 3.7-fold (p = 0.005), 4.4-fold (p = 0.004), 2.1-fold (p = 0.043) and 2.3-fold (p = 0.014), respectively. In intestinal tissue, SULT1B1 and SULT1D1 increased by 3.2-fold and 2.9-fold at 20 weeks, respectively (p > 0.05); and SULT1A1 and SULT1D1 increased by 2.0-fold (p >0.05) and 1.8-fold at 26 weeks (p = 0.024), respectively. In kidney tissue, SULT1D1 was upregulated by 3.0-fold at 26 weeks (p > 0.05).

Figure S3 shows some changes in DMEs with KRAS mutations in female KRAS mice. In liver tissue, UGT1A9 decreased by 2.3-fold at 5 weeks (p = 0.002); UGT2B1 decreased by 2.1-fold at 10 weeks (p = 0.004); CYP2C29, CYP2D22, CYP2E1, CYP3A11, CYP27A1, UGT1A1, UGT2A3, UGT2B5 and UGT2B1 were upregulated by 2.5- (p = 0.014), 1.7- (p = 0.018), 2.8- (p = 0.001), 2.6- (p = 0.001), 1.8- (p = 0.014), 2.5- (p = 0.008), 2.2- (p = 0.005), 2.3-fold (p = 0.014), and 2.4 folds (p = 0.002), respectively, at 15 weeks; SULT1A1 increased by 2.2 folds at 20 weeks (p > 0.05); and CYP3A11 decreased by 2.1-fold at 26 weeks (p = 0.033). In intestine tissue, SULT1B1 and SULT1D1 increased by 3.4-fold (p > 0.05) and 4.7-fold (p = 0.042), respectively, at 26 weeks. In kidney tissue, SULT1D1 increased by 1.9 folds at 26 weeks (p = 0.013).

Alterations in DME activities of DMEs by KRAS mutation

As shown in Fig. 3, the activity of CYP3A11 was significantly downregulated in the male KRAS mice with aging. Furthermore, there were significant differences between the WT and KRAS mice. UGT1A9 gradually declined in the male mice from 5 to 26 weeks. SULT1A1 and SULT1D1 displayed larger differences at 20 and 26 weeks in the male KRAS mice than in the WT mice. SULT1A1 increased by 2.2- (p >0.05) and 3.9-fold (p = 0.008), respectively. SULT1D1 was upregulated by 7.0- (p = 0.007) and 3.5-fold (p > 0.05), respectively. Figure S4 shows the activities in the female WT and KRAS mice at different ages. The activities of CYP2E1 and SULT1D1 displayed no significant differences in the WT and KRAS mice with increasing age. CYP3A11 displayed an increasing tendency from 5 to 26 weeks. The activity of UGT1A9 showed a significant decrease at 15, 20 and 26 weeks compared to that at 5 weeks. At 15 weeks, the activity of SULT1A1 was markedly different in the KRAS mice relative to the WT mice (p = 0.022).

Figure 3 Changes in the enzyme activity of (A) CYP2E1, (B) CYP3A11, (C) UGT1A9, (D) SULT1A1 and (E) SULT1D1 in liver tissues of the male KRAS and WT mice at different ages (n = 5).

The solid line represents male KRAS mice, and the dashed line represents male WT mice. Each data point is presented as the mean ± SD. For the comparison between KRAS and WT at the same age, the data were analyzed by independent sample t-tests (for normally distributed data) and Mann–Whitney U analysis (for non-normally distributed data). The symbol “*” indicates a significant difference between the male WT and KRAS mice at the same age, p < 0.05. For different ages compared to 5 weeks, the data were analyzed by one-way ANOVA (for normally distributed data) and Kruskal-Wallis H analysis (for non-normally distributed data). We adjusted the significance level α to 0.0125 according to the Bonferroni correction (0.05/4=0.0125). The symbols “A” and “a” indicate significant differences in the male WT and KRAS mice at 10, 15, 20 and 26 weeks relative to 5 weeks, p < 0.0125.

Figure 4 Correlation between the protein expression and activity of (A) CYP2E1, (B) CYP3A11, (C) UGT1A9, (D) SULT1A1 and (E) SULT1D1 in the liver tissue (n = 100).

The correlation between the protein expression and activity included 5, 10, 15, 20 and 26 weeks, which were analyzed together. DME in the liver was determined using an isotope label-free LC-MS/MS method. The enzyme activities of DMEs were measured using probe substrates. All measurements were performed in triplicate and the data are presented as the mean ± SD. Pearson product correlation and Spearman’s rank correlation were used to analyze the correlation. Regression line is shown for significant correlation at p < 0.05.

Correlation of protein content and enzyme activity of DMEs

The enzyme activities of CYP2E1, CYP3A11, UGT1A9, SULT1A1 and SULT1D1 were compared with their protein contents. The correlation analysis assessed the protein levels quantified by LC-MS/MS and the activities detected by the specific probes. As shown in Fig. 4, there was a good correlation between enzyme activity and protein content (CYP2E1, r2 = 0.54, p <0.001; UGT1A9, r2 = 0.55, p < 0.001; SULT1A1, r2 = 0.62, p < 0.001; SULT1D1, r2 = 0.89, p < 0.001). A poor correlation for CYP3A11 was observed in the mouse liver (r2 < 0.50, p > 0.05).

Protein expression profiles of AHR, FXR, PPARα (H-2) and PXR

To evaluate the protein levels of receptors, we dissected liver tissue from the KRAS mice. As shown in Fig. 5, the protein expression levels of AHR, FXR and PPARα (H-2) were downregulated by 40.22%, 20.90% and 26.76% in the livers of the male KRAS mice, respectively (p = 0.000, 0.035 and 0.005, respectively). Compared to the WT mice, the KRAS mice showed no significant difference in the protein amount of PXR (decreased by 13.52%, p = 0.109). In the female mice (Fig. S5), the protein expression levels of AHR, FXR, PPARα (H-2) and PXR were downregulated by 12.00%, 10.14%, 5.60% and 38.06% in the liver, respectively (p = 0.466, 0.442, 0.710 and 0.074, respectively).

Figure 5 Protein expression levels of AHR, FXR, PPARα and PXR in the male WT (n = 5) and KRAS mice (n = 4) at 26 weeks.

(A) The mprint of five proteins was represented and β-ACTIN was used as an internal control. (B) The data on protein expression levels was shown as a box chart. The data were analyzed by independent sample t-tests (for normally distributed data) and Mann–Whitney U analysis (for non-normally distributed data). The symbol “*” indicates a significance difference of protein expression level in the KRAS mice relative to that in the WT mice, p < 0.05.

Tissue distribution of DMEs

To evaluate the tissue distribution of DMEs, we present data for male mice at 10 weeks as an example. Figure 6 displays the distribution of DMEs in liver, intestine and kidney tissue. In liver, CYP2C29 >CYP2D22 >CYP3A11 ≈ CYP2E1 ≈ CYP1B1 >CYP7A1 >CYP27A1 ≈ CYP3A25; UGT2B5 >UGT2B1 >UGT1A6a >UGT1A1 >UGT2B34 ≈ UGT2B36 >UGT2A3 >UGT1A9 ≈ UGT1A5 >UGT1A2 (lower limit of quantification, LLOQ); SULT1A1 >SULT1D1 >SULT1B1 (LLOQ). The protein contents of UGT2B5, UGT2B1, UGT1A6a, CYP2C29, CYP2D22, UGT1A1 and SULT1A1 were the highest. In the intestine, CYP1B1 >CYP2D22 >CYP3A11 >CYP3A25 ≈CYP7A1 >CYP2E1/CYP2C29/CYP27A1 (LLOQ); UGT2B34 >UGT1A1 >UGT2A3 >UGT1A2/ UGT1A5/ UGT1A6a/UGT1A9/UGT2B1/UGT2B5/UGT2B36 (LLOQ); SULT1B1 >SULT1A1 >SULT1D1. The protein contents of CYP1B1, UGT2B34, SULT1B1, CYP2D22, CYP3A11, SULT1A1 and CYP3A25 were the highest. In the kidney, CYP1B1 >CYP2E1 >CYP7A1 >CYP2C29/CYP2D22/CYP3A11/CYP3A25/CYP27A1 (LLOQ). In the kidney, UGT1A2 was detected, but the other UGT isoforms were below the lower limit of quantification; for SULT isoforms, SULT1D1 was detected, but the others were all below the lower limit of quantification. SULT1D1, CYP1B1, CYP2E1, CYP7A1 and UGT1A2 had the highest protein expression levels.

Figure 6 Protein content of DMEs in different tissues from the KRAS and WT mice with different sexes at 10 weeks, n = 5.

All measurements were performed in triplicate and the data are presented as the mean ± SD. (A–B) Protein levels of eight CYP isoforms in different tissues from the KRAS and WT mice with different sexes. (C–D) Protein levels of 10 UGT isoforms in different tissues from the KRAS and WT mice with different sexes. (E–F) Protein levels of three SULT isoforms in different tissues from the KRAS and WT mice with different sexes.

Figure 7 Alterations in protein levels of eight CYPs, nine UGTs and two SULTs at different ages in the liver of male KRAS and WT mice, n = 5.

The dotted and solid lines represent the WT and KRAS mice, respectively. Each data point represents the mean ± SD. The data were analyzed by one-way ANOVA (for normally distributed data) and Kruskal–Wallis H analysis (for non-normally distributed data). We adjusted the significance level α to 0.0125 according to the Bonferroni correction (0.05/4=0.0125). The symbols “A” and “a” indicate significant differences in the male WT and KRAS mice at 10, 15, 20 and 26 weeks relative to 5 weeks, p < 0.0125.

DME variations based on sex

Figure 6 shows that the the sex-related changes in DMEs have a similar trend in both the WT and KRAS mice at 10 weeks of age. Therefore, we mainly discuss the differences in protein content in WT mice. In kidney tissue, CYP2E1 was male-predominant, while UGT1A2 was female-predominant. In liver tissue, the content of UGT2B1 was significantly higher in the male mice than in the female mice.

Figure 8 Alterations in protein levels of CYPs, UGTs and SULTs at different ages in the intestinal and kidney tissue of male KRAS and WT mice, n = 5.

A–K shows the DMEs expression in intestinal tissue. L-O shows the DMEs expression in kidney tissue. The dotted and solid lines represent the WT and KRAS mice, respectively. Each data point represents the mean ± SD. The data were analyzed by one-way ANOVA (for normally distributed data) and Kruskal–Wallis H analysis (for non-normally distributed data). We adjusted the significance level α to 0.0125 according to the Bonferroni correction (0.05/4=0.0125). The symbols “A” and “a” indicate significant differences in the male WT and KRAS mice at 10, 15, 20 and 26 weeks relative to 5 weeks, p < 0.0125.

Variations in DME protein content with increased age

In the liver, CYP7A1 increased with increasing age in the male mice (Fig. 7G); UGT1A9 showed a decreasing trend with increasing age (Fig. 7L). In the intestine, the CYP isoforms showed no significant changes at different ages (Figs. 8A–8E). UGT2B34 showed a decreasing trend in the male WT and KRAS mice with increasing age (Fig. 8H), and SULT1A1 showed an increasing trend with increasing age (Fig. 8I). In the kidney, CYP1B1 displayed a decreasing trend in the male WT and KRAS mice with increasing age (Fig. 8L). Figure S6 shows the changes of protein amount with aging in female WT and KRAS mice. In the liver, the protein amount of CYP2C29 decreased 2.9-fold at 15 weeks compared to that at 5 weeks in the WT mice. This pattern of changes did not appear in the KRAS mice. High individual differences in the protein amounts in female mice were observed. There are no significant differences at different ages.

Discussion

In this study, we systematically investigated the alterations of DMEs in KRAS mice of different ages and sexes, with the aim of providing a better explanation for the clinically observed variation in the efficacy and toxicity of anticancer drugs in KRAS-mutant lung cancer patients. Currently, limited information is available concerning the changes in DMEs in patients with KRAS mutant lung cancer.

The absolute protein contents of 21 metabolic enzymes in KRAS mice were simultaneously determined by the LC-MS/MS approach. In our study, the protein expression levels of CYP2C29 and CYP3A11 were significantly downregulated in the male KRAS mice at 26 weeks of age. A previous study indicated that hepatic DMEs are reduced during infection and inflammation in humans, rats, and mice (Moriya, Kataoka & Fujino, 2012). Similar results were also reported that a decrease in Cyp gene expression and enzymatic activity was observed in a dextran sulfate sodium (DSS)-induced mouse model of ulcerative colitis (Kusunoki et al., 2014). CYP2C29 is the major arachidonate CYP2C epoxygenase in mice (Sodhi et al., 2009). The decreased expression of CYP2C29 is closely related to the occurrence and development of inflammation. Related studies have shown that this decreased expression may be triggered by an increased production of inflammatory cytokines (Kusunoki et al., 2014). CYP3A11 plays a vital role in the metabolism of various clinical anticancer drugs, such as erlotinib, cisplatin, sorafenib. The drug concentration in serum would change accordingly with enzymatic expression. The declining expression of CYP3A11 in the KRAS mice may cause some differences in efficacy or even side effects. With respect to the SULT family, the protein expression and activities of SULT1A1 and SULT1D1 were upregulated in the male KRAS mice at 20 and 26 weeks. A number of studies have been conducted on SULT in different cancers, but many conflicting outcomes have been reported (Jiang et al., 2010). Some authors showed a potential association between SULT1A1 polymorphisms and breast cancer, but inconsistent results also exist (Jiang et al., 2010). Relevant studies have reported that SULT1A3 may be a diagnostic marker for osteosarcoma, and SULT1A3 protein upregulation is closely related to the occurrence and development of cancer (Chen et al., 2014a; Chen et al., 2014b; Xie et al., 2017). SULT1D1 is a pseudogene in humans, Sult1d1 encodes protein expression in mice, and its functions are similar to those of human SULT1A3 (Wong et al., 2010). Our results also revealed increased protein expression of SULT1D1 in male mice after KRAS mutation. This finding is beneficial to explain the metabolic characteristics of SULT1D1-metabolized drugs in KRAS mice. The expression of SULT1A3 should be further explored in KRAS-mutant lung cancer patients.

To further explore the changes in enzymatic activity, we used specific probe substrates to detect the enzymatic status in the KRAS mice. In our present study, we found the CYP2C29/CYP3A11/SULT1A1/SULT1D1 displayed significant changes in protein expression in the liver of male WT and KRAS mice. Therefore, we are intended to research their activities in the liver tissue. We failed to find an authoritative and specific probe to study the activity of CYP2C29. Chlorzoxazone and Propofol are usually used as specific substrates to study the activities of CYP2E1 and UGT1A9. Their good correlation between protein expression and activities indicated the protein quantification results are credible. Therefore, we select them for the enzyme activity test. Notably, SULT1A1 and SULT1D1 activity was upregulated at 20 and 26 weeks in the male KRAS mice (Fig. 3 and Fig. S4). This result was consistent with their protein expression levels. In this context, a high degree of correlation was observed between the enzymatic activity and protein level (Fig. 4). For the poor correlation between the enzyme activity and protein level of CYP3A11, the nonspecificity of the substrate may be a possible reason. The FDA (USA) reported that testosterone was metabolized by CYP3A4 and CYP3A5. In addition, the protein structure could affect the activity. CYP3A4 showed a significant genetic polymorphism in individuals, causing a flexible three-dimensional structure of CYP3A4 (Werk & Cascorbi, 2014). Moreover, the genetic polymorphism of CYP2D6 (ortholog of CYP2D22 in mice) could induce variations in the expression or function of CYP3A4 (Werk & Cascorbi, 2014). Generally, these findings indicate that the protein expression levels of some DMEs could be applied to forecast the enzymatic activities regarding drug metabolism. Changes in the ability of DMEs to metabolize drugs could lead to differences in the ADME properties of drugs, thereby affecting drug efficacy and toxicity in the body.

The expression of DMEs is regulated by the binding of xenobiotics to receptors, such as the aryl hydrocarbon receptor (AHR), the murine pregnane X receptor (PXR), peroxisome proliferator-activated receptor (PPARα) and bile acid receptor (FXR) (Anakk et al., 2003; Handschin & Meyer, 2003; Honkakoski & Negishi, 2000). The decrease in receptor levels may contribute to the emergence of changes in DME expression (Anakk et al., 2003; Li et al., 2009; Moscovitz et al., 2018). Moreover, some reports have suggested that disease status (e.g., cancer and inflammation) can affect the expression and activity of DMEs via specific receptors (Lamba, Jia & Liang, 2016; Chen et al., 2014a; Chen et al., 2014b; Schröder et al., 2011). Therefore, we further studied the changes in receptor expression after KRAS mutation. In our study, the protein expression levels of AHR, FXR and PPARα were downregulated in the livers of the male KRAS mice compared to the WT mice at 26 weeks. This phenomenon was not significant in the female KRAS mice. Major xenobiotic-sensing transcription factors, such as AHR and PXR, are involved in the regulation of the protein expression of DMEs. Related reports revealed that most core DMEs were positively correlated with AHR, PXR and PPARα, and their protein expression was downregulated in nearly 50% of the patients with hepatocellular carcinoma (HCC) (Chen et al., 2014a; Chen et al., 2014b; Hu et al., 2018; Zhong et al., 2016). Activation of AHR could induce the upregulation of Cyp1a/3a/Ugt1a1 mRNA expression, which would therefore not occur in Ahr-null mice (Klaassen & Slitt, 2005; Nakajima, Masashi & Tsuyoshi, 2003). PPARα and PXR were implicated in the regulation of CYP3A/4A/1A1/2B6/2C8/2C9/2C19/UGT1A1 induction (Moscovitz et al., 2018). FXR, an important regulator of lipid and glucose homeostasis, is involved in the expression of CYP7A1 and CYP27A1 (Sánchez, 2018). Hence, in our study, we speculate that these variations in DME expression may be regulated by decreased receptors of AHR, FXR and PPARα.

For sex-difference, CYP2E1 showed significant male-specificity in kidney tissue. CYP2E1 mediates the metabolism of many low molecular weight organic compounds (such as ethanol and acetone) and some drugs (such as p-nitrophenol, caffeine, chlorzoxazone, etc.) (Löfgren et al., 2004; Zuber, Anzenbacherová & Anzenbacher, 2002). Therefore, in regard to the intake of these exogenous substances, we should consider the effects related to sex differences in patients. For UGT enzymes, UGT2B1 exhibited male-predominant expression in the liver tissue. Conversely, UGT1A1 and UGT1A5 expression in the liver and UGT1A2 in the kidney are female-predominant, whereas UGT2B1 exhibited male-predominant expression in liver tissue. These results are consistent with previous reports (Buckley & Klaassen, 2007; Buckley & Klaassen, 2009). The female-predominant UGT1A1 expression accounts for the higher bilirubin-UGT activity in females. UGT1A and UGT2B are the primary families of UGT enzymes, involved in the inactivation of >30% of drugs currently used in the clinic (Guillemette, 2014). These sex-specific expressions may be crucial in understanding the mechanisms by which many drugs display variations in metabolism and clearance.

For age-related difference, except for UGT1A9, the majority of DMEs showed no significant changes from 5 to 26 weeks of age in female and male WT and KRAS mice. UGT1A9, major UGT isoforms expressed in the liver (∼6% of hepatic UGT expression), is responsible for the glucuronidation of multiple endogenous substances (e.g., thyroid hormones) and drugs (e.g., acetaminophen and propofol) (Cho et al., 2016). The activity and protein expression of UGT1A9 appeared to decrease in the liver of female and male WT mice. Related studies indicated that UGT1A9 activity showed a downward trend from 6 weeks to 52 weeks in mice with a FVB background (Zheng et al., 2018). Therefore, the appropriate dosage should be considered when patients of different ages are prescribed drugs metabolized by UGT1A9.

For tissue-related differences, abundant CYP enzymes were expressed in the liver, predominantly CYP2D22, CYP2C29, CYP2E1 and CYP3A11 (Figs. 6A–6B). A previous studies also demonstrated that the protein contents of these isoforms were high in the liver (Chen et al., 2017; Gröer et al., 2014). In the intestine, the CYP1B1, CYP2D22, CYP2E1 and CYP3A11 protein levels were significantly higher than the levels of other proteins. Mouse phase I enzymes (CYP2D22, CYP2E1 and CYP3A11) are orthologs of the corresponding human enzymes (CYP2D6, CYP2E1 and CYP3A4), in charge of major phase I-dependent metabolism in marketed drugs (Jin, Tawa & Wallqvist, 2013; Ruaño et al., 2012). Hence, optimal drug administration routes should be considered when these enzymes are involved in the inactivation or activation of drugs.

Conclusion

Taken together, our data showed significant decrease in CYP3A11 and CYP2C29, but an increase in SULT1A1 and SULT1D1 in the KRAS mice at 26 weeks. These DMEs all participate in the metabolism of drugs. Therefore, we hope that these results could provide useful guidance or a theoretical basis for further drug research and implementation.

Supplemental Information

Supplemental Information 1 Dynamic MRM chromatograms of 21 subtypes of DMEs for the LC-MS/MS methods

(A) the standards of CYPs. (B) the standards of UGTs and SULTs.

Click here for additional data file.

Supplemental Information 2 H&E staining of the lung tissue from the male WT and KRAS mice at 5, 10, 15, 20 and 26 weeks

The arrows partly pointed out the hyperproliferative lung cells in KRAS mice.

Click here for additional data file.

Supplemental Information 3 The relative expression levels of DMEs in liver, intestine and kidney tissue in the female mice at different ages

Protein levels in the female WT mice ( n = 5 ) were normalized to those in the female KRAS mice ( n = 5 ). The data were analyzed by independent sample t tests (for normally distributed data) and Mann-Whitney U analysis (for non-normally distributed data). The symbol “*” indicates a displayed significant difference between the male WT and KRAS mice at the same age, p < 0.05.

Click here for additional data file.

Supplemental Information 4 Changes in the enzyme activity of CYP2E1, CYP3A11, UGT1A9, SULT1A1 and SULT1D1 in liver tissue of the female KRAS and WT mice at different ages

Each data point is presented as the mean ±SD. For the compare between KRAS and WT at the same age, the data were analyzed by independent sample t tests (for normally distributed data) and Mann–Whitney U analysis (for non-normally distributed data). The symbol “*” indicates a significant difference between the female WT and KRAS mice at the same age, p < 0.05. For different age compared to 5 weeks, the data were analyzed by one-way ANOVA (for normally distributed data) and Kruskal–Wallis H analysis (for non-normally distributed data). We adjusted the significance level α to 0.0125 according to the Bonferroni correction (0.05/4=0.0125). The symbols “A” and “a” indicate significant differences in the female WT and KRAS mice at 10, 15, 20 and 26 weeks relative to 5 weeks, p < 0.0125.

Click here for additional data file.

Supplemental Information 5 Protein expression levels of AHR, FXR, PPAR α and PXR in the female WT ( n = 5 ) and KRAS mice ( n = 3) at 26 weeks

(A) The mprint of five proteins was represented and β-ACTIN was used as an internal control. (B) The data on protein expression levels was shown as a box chart. The data were analyzed by independent sample t tests (for normally distributed data) and Mann–Whitney U analysis (for non-normally distributed data). The symbol “*” indicates a significance difference of protein expression levels in the KRAS mice relative to that in the WT mice, p < 0.05.

Click here for additional data file.

Supplemental Information 6 Alterations in protein levels of 8 CYPs, 9 UGTs and 2 SULTs at different ages in the liver of female KRAS and WT mice, n = 5

The dotted and solid lines represent the WT and KRAS mice, respectively. Each data point represents the mean ±SD. The data were analyzed by one-way ANOVA (for normally distributed data) and Kruskal-Wallis H analysis (for non-normally distributed data). We adjusted the significance level α to 0.0125 according to the Bonferroni correction (0.05/4=0.0125). The symbols “A” and “a” indicate significant differences in the male WT and KRAS mice at 10, 15, 20 and 26 weeks relative to 5 weeks, p < 0.0125.

Click here for additional data file.

Supplemental Information 7 Alterations in protein levels of CYPs, UGTs and SULTs at different ages in the intestinal and kidney tissue of female KRAS and WT mice, n = 5

A-K shows the DMEs expression in intestinal tissue. L-O shows the DMEs expression in kidney tissue. The dotted and solid lines represent the WT and KRAS mice, respectively. Each data point represents the mean ±SD. The data were analyzed by one-way ANOVA (for normally distributed data) and Kruskal-Wallis H analysis (for non-normally distributed data). We adjusted the significance level α to 0.0125 according to the Bonferroni correction (0.05/4=0.0125). The symbols “A” and “a” indicate significant differences in the male WT and KRAS mice at 10, 15, 20 and 26 weeks relative to 5 weeks, p < 0.0125.

Click here for additional data file.

Supplemental Information 8 Supporting information

Click here for additional data file.

Supplemental Information 9 Raw data

Click here for additional data file.

We thank Tongmeng Yan of State Key Laboratory of Quality Research in Chinese Medicine, Macau University of Science and Technology, Macau (SAR), China for the method of S9 sample preparation. We thank Guoxin Huang of State Key Laboratory of Quality Research in Chinese Medicine, Macau University of Science and Technology, Macau (SAR), China for checking English language problems.

Additional Information and Declarations

Competing Interests

Author Contributions

Animal Ethics

Data Availability

The authors declare there are no competing interests.

Xiaoyan Li conceived and designed the experiments, performed the experiments, analyzed the data, prepared figures and/or tables, authored or reviewed drafts of the paper, and approved the final draft.

Yiyan Lu performed the experiments, prepared figures and/or tables, and approved the final draft.

Xiaojun Ou and Xiaoxiao Qi performed the experiments, authored or reviewed drafts of the paper, and approved the final draft.

Sijing Zeng analyzed the data, prepared figures and/or tables, and approved the final draft.

Ying Wang analyzed the data, authored or reviewed drafts of the paper, and approved the final draft.

Lijun Zhu and Zhongqiu Liu conceived and designed the experiments, authored or reviewed drafts of the paper, and approved the final draft.

The following information was supplied relating to ethical approvals (i.e., approving body and any reference numbers):

The Institutional Animal Care and Use Committee of the International Institute for Translational Chinese Medicine provided full approval for this research (IITCM_20171105).

The following information was supplied regarding data availability:

The raw measurements are available in the Supplemental Files.

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
