# Peer review of "Changes and sex- and age-related differences in the expression of drug metabolizing enzymes in a KRAS-mutant mouse model of lung cancer"

_PeerJ, doi:10.7717/peerj.10182_

## Round 0.1 · original submission · Major Revisions

Our reviewers request several changes and clarifications. Please address them thoroughly. I noticed that you do not seem to have performed a multiple-comparison correction to your p-values. It is therefore likely that several of your significant results are simply due to the extremely large number of comparisons performed. Please apply a proper correction for that.

·

Basic reporting

Age, sex and disease-state could have an impact on the expression of drug metabolizing enzymes, which could then contribute to the differences in drug efficacy and toxicity. In this study, Li et al. attempt to profile the effect of sex, age and disease-state on the expression of DMEs in a lung cancer model. While it is important to profile the age-and sex- related differences in DME expression in a lung tumor model to predict adverse drug reactions or lack of response of chemotherapeutic agents, the study presents numerous limitations. Incoherent language and grammatical errors throughout the paper make it very difficult to read. I advise the authors to kindly revise the language of the manuscript thoroughly to make it clearer. The references provided were inadequate and had several errors. Few references were cited multiple times and most of the papers cited were review articles and not original papers. It was unclear whether independent experiments were performed to validate the study. While they show differences in DME levels with respect to age, sex and disease state, they fail to discuss any possible hypothesis for these observations.

Please see my specific comments below:

Line 51-52 “Therefore, it is necessary to investigate the expression of DMEs for drug safety and availability.” This sentence is abrupt and misleading. Please amend the sentence to reflect that you are specifically investigating the age and sex-related differences in DME expression levels in a lung cancer model.

Line 53-60. Please revise the language and sentence structure in this paragraph. Provide justification to choosing KRAS model to study differences in DME expression in a lung cancer context in a concise manner.

Line72. Please change the sentence to “Sex and age are important factors influencing the expression level of DMEs”. Include references.

Please include references for the following lines: Line 65, 67, 72

Line 76. Please be more specific about which enzyme levels/ oral drugs you are referring to. Also, please find an appropriate reference for this. Waxman and Holloway 2009 is a review paper and it does not conclude that there is an increase in clearance rate of oral drugs by 17-26% in females. Cite original and not reviews articles whenever possible.

Line 91. Change the sentence to “.. can increase the sensitivity and throughput..”

Experimental design

Methods:

Line 112. Correct typo in C57 BL/6

Line 168-168. Change the sentence to “… were diluted to a recommended dilution of 1:1000 with 5% milk..”

Line 130. What microscope was used for the histopathological analysis?

Please include details the following details in your methods/results section:
You report average of technical replicates in most of the figures. How many independent experiments were performed? This is very important to determine whether the data was reproducible.

Please specify the number of animals used per group in EACH study.

Results:
Fig 3 and Supplemental Fig 4- Please mention the sex of the mice on the graph.

Supplemental Fig 3. Why is there an increase in many of the enzyme expression levels in female KRAS mice at week 15 in the liver? (Not seen in male mice at the same time point). If this data is reliable, please discuss the possible reasons for this.

Fig 4. What are the possible reasons for the enzyme activity of Cyp3a11 not correlating with the protein levels in the liver? Your attempt to discuss this in Line 301 is not satisfactory.

Fig. 5 – It is not acceptable to have so much variation in your beta –actin bands (loading control) across the samples. Please label your figures appropriately. Which lanes correspond to WT and KRAS mice? What about the receptor expression in female WT and KRAS mice?

Fig 6- Why did you choose 10 weeks as your time point for this study?
Line 248 “ the differences in DME content based on sex are similar in KRAS mice and WT mice” This statement is not clear. Are you implying that sex-related changes in DME enzymes have a similar trend both in WT and KRAS mice? While this is true at 10 weeks, you see different trends in male and female KRAS mice after week 15 (Fig 3 and supplemental Fig 4) and this needs to be discussed. According to the literature, Cyp3A4 (human ortholog of Cyp3a11) is expressed at higher level in females than males. Can you discuss why you do not see a similar trend (at least in the WT mice at week 10) in your study?

Validity of the findings

Discussion:

Line 272. You statement does not reflect that the downregulation of Cyp2c29 and Cyp3a11 was observed only in male and not female KRAS mice at 26 weeks.

Line 277, 310 and 50 -you cite Kusunoki 2015. Is it Kusunoki 2015 or 2014? In Line 281 you refer to the same paper as Kusunoki 2014. Please double check ALL your references.

Line 285-286. Again, upregulation in Sult1a1 and Sult1d1 enzyme levels was seen in only male but not female KRAS mice after week 20.

What is your hypothesis for altered DME expression is KRAS mice?

How do you explain the sex-related differences in expression of hepatic enzymes in older KRAS mice? Do hormones have a role to play in the expression of DME?

I do not agree with the conclusive remarks of this study. This study attempts to profile sex-and age-related differences in DME expression in KRAS mice. However, female KRAS mice do not show the same age-related changes in DME expression as the male KRAS mice. This needs to be highlighted and discussed in detail. All the conclusions of altered DME expression in KRAS mice were drawn from the data with the male mice (with the data from female mice merely provided as supplemental data).

Reviewer 2 ·

Basic reporting

The article is largely well written and structured. However, it could use a proper revision for grammar and sentence structure in a few places, for example lines 42, 43, 53, 73, 74, 83, 91 etc.

The results and conclusions are within the stated hypotheses and do no make any exaggerated claims.

Points to address:
- please state what is S9 fraction
- some nomenclature is inconsistent. For example, pay attention to case in gene and protein names. in vivo/ in vitro should be consistently italicized
- Fig 2b, what is the x-axis? The figure is not clear
-Fig 5a, what are the different lanes in the western blot? Actin expression is not consistent. Is the higher expression normalized with other genes in that lane?

Experimental design

The question is well defined and the authors present relevant data to address the aims of the study.

Validity of the findings

The conclusions are clear and discussion of the data has been made in a sufficient way with proper links to existing literature.

Reviewer 3 ·

Basic reporting

-

Experimental design

-

Validity of the findings

-

Additional comments

In this work by Li et al. the authors investigate the sex- and age-dependent differences in the expression of 21 drug-metabolizing enzymes in WT and KRAS mice. The findings are of interest; nonetheless, I do have few questions and suggestions regarding the study to clarify these points.
1. There are various repetitions in the text-lines 47, 66, 73, 76 and lines 51, 78
2. Please provide the references for the statements in lines 54, 56, 57, 61, 65, 72, 87
3. The language in the present manuscript requires major improvement – it is full of errors and typos. There are many grammatical, linguistic and spelling mistakes throughout the manuscript.
Some examples where language can be improved include lines 115, 116, 119, 121, 122, 129, 132 etc
4. Please add scale bars to the images - Fig. 1 and Supplemental Fig. 2
5. It would help if the histopathology described (line 189-190) is marked on the images (fig. 1b and supplemental fig. 2) with arrows.
6. Did the authors see any difference in the size and the number of lung nodules across different age groups in KRAS mice?
7. Please provide a more detailed description of the results part.
For example, the description of the PLS-DA analysis should be improved (line 191-193)
-line 223-226 please provide a detailed explanation
8. The authors refer to the expression of Sult1a1 (20 weeks) and Sult1a1/Sult1d1 (26 weeks) in the liver as significant (p< 0.05); however, the asterisk is missing from the figure (Fig. 2b).
9. Symbols for genes should be italicized, for example Cyp3a11
10. Please explain, in the results part, based on what criteria were only Cyp2e1, Cyp3a11, Ugt1a9 and Sult1a1 selected for the enzyme activity - Fig. 3?
11. Fig. 5, are the first 5 wells for WT mice? Please indicate that on the figure
- It is not clear, does Fig. 5b represent a normalization to actin?
- Please explain the choice of these particular receptors.
- Actin is unequal between samples, even in the same group, I would suggest
to use another loading control or to re-do the blot.
- The authors did not specify which age group was used for WB analysis.
12. It would be interesting to check if there is any difference in the protein
abundance of these receptors in the female mice.
13. In the discussion, authors use couple of sentences to discuss the major finding-sex- age- and tissue-related differences in the expression of DME. Please elaborate it more and include a paragraph illustrating a relevance of the main finding.

---

## Round 0.2 · Minor Revisions

Reviewer #3 still has some concerns. I hope they will not be hard to address.

·

Basic reporting

The authors have addressed the questions satisfactorily. The language has been improved considerably, with more details added to the methods section. I was satisfied with the changes made to the discussion as well.

Experimental design

No comments

Validity of the findings

No comments

Reviewer 2 ·

Basic reporting

The study presented here is much clearer and polished compared to the initial submission. The authors have made the changes to convert it into a more professional manuscript. The references are improved and figures have been made clearer. My comments on the initial submission have been addressed satisfactorily.

Experimental design

No comment

Validity of the findings

No comment

Additional comments

I wish you the best for your future endeavors

Reviewer 3 ·

Basic reporting

-

Experimental design

-

Validity of the findings

-

Additional comments

The authors addressed all concerns raised by the reviewers and the manuscript has been significantly improved. However, there are some points which still need further clarification;
1. Please correct
- line 59: change ‘’detoxifivcation’’ to detoxification
- line 167: change the sentence to ‘’before the experiment, animals were fasted overnight but allowed free access to water’’.
- Line 168: change the sentence to ‘’all procedures were performed…’’
- Line 180: change ‘’fixated’’ to fixed
- Line 224: change the sentence to ‘’Then, the solution was vortexed and thereafter centrifuged for…’’
- Line 226: change the sentence to ‘’ the enzyme activity was measured from 2 independent experiments’’
- Line 249: add ‘’was performed from 2 independent experiments’’

2. Line 169: it is not clear what do the authors mean by: ‘’at the conclusion of the experiment, the animals bodies were harmlessly treated by professionals’’
3. Line 199, the authors state that 10 isoforms of UGTs were analysed, however they show only 9 isoforms in Fig. 2 (F, G, H), UGT1A2 is missing
4. Line 269, the authors mention that cell membranes were broken in H&E images, however this is not clear from the image, please clarify or mark it on the image.
5. Please mention in the results part, for Fig. 2, what do the arrows correspond for
6. Authors state that the cancer stage of 20-week old KRAS mice corresponds to the advanced stage of human lung cancer. Could you please explain why did you then choose 26-week old mice mice for your experiments (western blot, histology), taking into consideration that these mice live 28 weeks and as Fig. 1 in the rebuttal letter shows, it is already difficult to distinguish lung morphology in mice at 26-weeks of age. And how does this age group (26-weeks) correlate to cancer stage in humans?
7. Figure 4, did the authors correlate the enzyme activities of 26-week old mice? Please mention the age in the figure legend
8. I would suggest to add to the main manuscript the explanation from the rebuttal letter, (Reviewer 3- question 10) of the criteria for selection Cyp2e1, Cyp3a11, Ugt1a9 and Sult1a1 for the enzyme activity - Fig. 3
‘’In our present study, we found the CYP2C29/CYP3A11/SULT1A1/SULT1D1 displayed significant changes in protein expression in the liver of male WT and KRAS mice. Therefore, we are intended to research their activities in the liver tissue. We failed to find an authoritative and specific probe to study the activity of Cyp2c29. Chlorzoxazone and Propofol are usually used as specific substrates to study the activities of Cyp2e1 and Ugt1a9. Their good correlation between protein expression and activities indicated the protein quantification results are credible. Therefore, we select them for the enzyme activity test, intending to show a good correlation between protein expression and activity’’.

---

## Round 0.3 · Minor Revisions

I am afraid that I seem to have found some discrepancies in different versions of Fig. 3: for example, in the first version, no significant differences were found in UGT1A9 at 20w, but upon applying the Bonferroni correction significant differences were found between KRAS and WT, as well as between WT 20w and WT 5w. This does not make sense, and I think some labeling mistakes were introduced. To clarify matters, please double-check every comparison and add the values of standard deviation and corresponding test statistics to the enzyme raw data in the Supporting information files.

Reviewer 3 ·

Basic reporting

The authors have addressed all raised points.
I would only ask the authors to rewrite/rephrase the sentence (line 70-71) using correct English.

Experimental design

-

Validity of the findings

-

Additional comments

-

---

## Round 0.4 · accepted · Accept

Thank you for clarifying the remaining issues. I look forward to seeing your manuscript in print!